# A Novel Kernel Sparse Coding Method with A Two-stage Acceleration Strategy

## Abstract

Sparse coding aims to exploit the latent linear structure of the input data, transforming dense data into sparse data, thereby improving data processing efficiency. However, many real-word signals cannot be expressed linearly, rendering the traditional sparse coding algorithms ineffective. One potential solution is to expand the dimensions of data. In this paper, we verify that the feature mapping of Radial Basis Function (RBF) kernel contains infinite dimensional information, and it does not significantly increase the computational complexity. Based on this, we propose to explore the $l_1$-norm regularization sparse coding method with RBF kernel, and provides a solution with convergence guarantees by leveraging the principle of coordinate descent. Additionally, to accelerate the optimization process, we introduce a novel two-stage acceleration strategy, based on theoretical analysis and empirical observations. Experimental results demonstrate that the two-stage acceleration strategy can reduce processing time by up to 90%. Furthermore, when the data size is compressed to about 2% of its original scale, the NMAE metric of the proposed method reaches as low as 0.0824 to 0.2195, achieving a significant improvement of up to 47% compared to traditional linear sparse coding methods and 36% compared to other kernel sparse coding techniques.

## 1 Introduction

Sparse coding, known for its efficiency in transforming data into a sparse and interpretable format, posits that data in the input space possesses inherent features (Pati et al., 1993; Bao et al., 2014; Lee et al., 2006; Kim, 2014). Within mathematical models, these intrinsic features are typically modeled as a set of basic atoms or dictionary atoms. As illustrated in Fig. 1, these atoms can be concatenated into a dictionary matrix $\mathbf{X}$. Once the input data $\mathbf{y}$ can be expressed as a linear combination of a few dictionary atoms (e.g., Atom 2 and Atom 6), it can be encoded as sparse weights data $\mathbf{w}$, where the majority of weights (e.g., $w_j$ for $j = 1, 3, 4, 5, 7, 8$) are zeros and only a few weights (e.g., $w_2$, $w_6$) are nonzeros. Owing to the unique advantages of sparse data, sparse coding have the potential capability to handle the issues of data compression (Ramabadran & Sinha, 1994; Watkins et al., 2018; Li et al., 2016; Wang et al., 2016), efficient computation (Schütze et al., 2016; Chalk et al., 2018; Bengio et al., 2009; Schütze et al., 2016), feature analysis (Yutani et al., 2022; Shi et al., 2021; Tong et al., 2019), denoising (Liu et al., 2019; Wang et al., 2020; Lu et al., 2015).

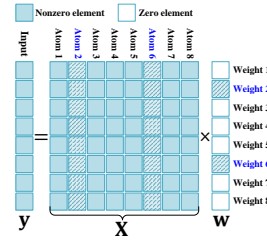

Figure 1: Sparse coding.

Within the aforementioned sparse coding framework, the input data $\mathbf{y}$ can be sparsely encoded into $\mathbf{w}$ because it can be linearly reconstructed by a combination of several dictionary atoms. It is worth noting that the dictionary atoms are typically learned directly from the input space, leveraging the fundamental assumption that the input data possesses inherent linear structures. However, (Yang et al., 2016) has shown that many practical signals do not possess this characteristic, limiting the effectiveness of traditional linear sparse coding methods in such scenarios. Therefore, seeking methods to effectively address the sparse coding of complex data is a highly meaningful topic.

**Motivations.** Recent studies show that a transformation of low-dimensional data into a higher-dimensional space, may enable data that is inherently nonlinearly separable in its original form to become linearly distinguishable within the expanded feature space (Schölkopf et al., 1997). For

example, as shown in Fig. 2(a), there is no straight line that can separate the green data points from the red data points. However, if we map each two dimensional data point $(x, y)$ into a three dimensional data point $(x, y, xy)$, as shown in Fig. 2(b), there will be numerous planes that can separate the green data points from the red data points. Inspired by this, increasing the dimensionality of input data may offer promising prospects for sparse coding problems involving data with complex internal structures.

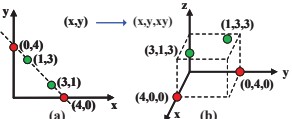

Figure 2: Transforming linearly inseparable data into linearly separable data.

While increasing the dimensionality of input data can enhance its discriminability, facilitating sparse coding analysis, it also inevitably introduce problems, including indeterminate forms and scales of data dimensionality enhancement, and high computational complexity. It is known that kernel trick provides an effective way to expand data dimensions with reducing computational complexity, and there are various categories of kernel functions (Gao et al., 2010). To select a good kernel function, we must be aware that the richer the dimensional information contained in the feature map, the better the performance of the corresponding kernel function. Therefore, this article delves into the specific form of feature mapping corresponding to the Radial Basis Function (RBF) kernel and analyzes its potential to demonstrate great performance. Ultimately, we solidify our decision to employ the RBF kernel-based approach as a means to tackle the challenge of enhancing sparse coding efficiency.

**Contributions.** The main contributions of this work include:
• We verify that the feature mapping of the Radial Basis Function (RBF) kernel contains infinite dimensional information, and it does not significantly increase the computational complexity. Based on this, we explore the $l_1$-norm regularized sparse coding method with RBF kernel, and provides a solution with convergence guarantees by leveraging the principle of coordinate descent.
• To reduce the computational complexity of the standard kernel sparse coding method based on coordinate descent, a novel two-stage acceleration strategy is introduced. This strategy focuses on updating the initially predicted nonzero weight factors, and effectively predicts zero weight factors to skip the computation of the corresponding intermediate variables.
• Our experiments demonstrate that the proposed two-stage acceleration strategy significantly reduces processing time by up to 90%. Additionally, our method outperforms both traditional linear sparse coding and other kernel sparse coding methods. Specifically, when the data size is compressed to about 2% of its original scale, the NMAE metric of the proposed method reaches as low as 0.0824 to 0.2195, achieving a significant improvement of up to 47% compared to traditional linear sparse coding methods and 36% compared to other kernel sparse coding techniques.

## 2 PRELIMINARY AND KERNEL TRICK

**Problem:** In practice, the effective implementation of sparse coding relies on a dictionary matrix $\mathbf{X} = [\mathbf{x}_1, \cdots, \mathbf{x}_p] \in R^{n \times p}$ that preserves the potential features of the input space, which should be learned in advance from the dense historical data $\mathbf{Y} \in R^{n \times m}$ as:

$$\min_{\mathbf{X}, \mathbf{W}} \frac{1}{2} \|\mathbf{Y} - \mathbf{X}\mathbf{W}\|_F^2 + \lambda \sum_{j=1}^{m} \|\mathbf{w}_j\|_1, \tag{1}$$

where $\mathbf{W} \in R^{p \times m}$ is the sparse coding matrix, with the majority of elements being zero, and only a minority of elements being nonzero. $\mathbf{w}_j$ is the $j$th column of $\mathbf{W}$. The first term for (1) is the approximation error, and the second term with penalty parameter $\lambda$ is used to force the columns of matrix $\mathbf{W}$ to be sparse.

Essentially, optimization problem (1) focuses on identifying a dictionary matrix $\mathbf{X}$ within the original data space, enabling the transformation of $\mathbf{Y}$ into a sparse matrix $\mathbf{W}$, based on the assumption that each column in $\mathbf{Y}$ can be expressed as a linear combination of a few atoms in $\mathbf{X}$. Since $\mathbf{W}$ contains significantly less amount of data than $\mathbf{Y}$, it can be seen as a sparse form of input data $\mathbf{Y}$. **However, when the input data $\mathbf{Y}$ does not have enough inherent linear structure, achieving a sparse linear approximation of $\mathbf{Y}$ with high precision proves challenging.**

For simplicity, we abbreviate "Number of Nonzero Elements" as "NNZE" in this paper. As shown in Example 1, the input data $\mathbf{Y}$ contains 10 data points uniformly and randomly sampled from the unit sphere. Assuming the dictionary matrix has 6 atoms, performing sparse coding and dictionary learning on $\mathbf{Y}$ in the original input space, with the constraint that the approximation error

**Example 1:** Comparison of SCDL in different data spaces

SCDL in the original space:

$$\mathbf{Y} = \begin{bmatrix} -0.9803 & 0.3126 & 0.7473 & -0.0897 & 0.5752 & 0.6247 & 0.3181 & -0.3795 & 0.6192 & -0.9283 \\ -0.1677 & 0.6755 & -0.5517 & 0.4611 & 0.7826 & 0.7287 & -0.8019 & 0.9212 & 0.5738 & -0.3354 \\ 0.1041 & 0.6679 & 0.3704 & 0.8828 & 0.2382 & 0.2807 & 0.5057 & -0.0854 & 0.5360 & -0.1602 \end{bmatrix}$$

NNZE(Y) = 30

$\widetilde{\mathbf{y}}_i = \left[ y_{1i}, y_{2i}, y_{3i}, y_{1i}^2, y_{2i}^2, y_{3i}^2, y_{1i}y_{2i}, y_{1i}y_{3i}, y_{2i}y_{3i} \right]^T, i = 1,2,\cdots,10$

SCDL O(180)

$$\mathbf{X} = \begin{bmatrix} 0.9619 & 0.1807 & 0.5859 & 0.1647 & 0.7811 & -0.0869 \\ 0.2520 & 0.9209 & -0.4397 & 0.9862 & 0.5918 & 0.4591 \\ 0.1058 & 0.3454 & 0.6807 & -0.0188 & 0.1991 & 0.8841 \end{bmatrix}$$

$$\mathbf{W} = \begin{bmatrix} -0.9507 & 0.0800 & 0.5077 & 0 & 0.3530 & 0.4045 & 0 & -0.2138 & 0.4454 & -0.8144 \\ 0 & 0.4326 & -0.1762 & 0 & 0.5106 & 0.5227 & 0 & 0.4501 & 0.2939 & -0.0189 \\ 0 & 0.1085 & 0.5508 & 0 & 0 & 0 & 0.6967 & -0.4769 & 0.0938 & -0.0053 \\ 0 & 0 & -0.2685 & 0 & 0.1304 & 0.0270 & -0.4993 & 0.2872 & 0 & 0 \\ -0.0450 & 0.1605 & 0 & 0 & 0.1464 & 0.1704 & 0 & 0 & 0.1329 & -0.1709 \\ 0.2278 & 0.4466 & 0 & 0.9900 & 0 & 0.0194 & 0.0075 & 0.1236 & 0.3247 & -0.0217 \end{bmatrix}$$

NNZE(W) = 40 > NNZE(Y)

SCDL in the high-dimensional feature space:

$$\widetilde{\mathbf{Y}} = \begin{bmatrix} -0.9803 & 0.3126 & 0.7473 & -0.0897 & 0.5752 & 0.6247 & 0.3181 & -0.3795 & 0.6192 & -0.9283 \\ -0.1677 & 0.6755 & -0.5517 & 0.4611 & 0.7826 & 0.7287 & -0.8019 & 0.9212 & 0.5738 & -0.3354 \\ 0.1041 & 0.6679 & 0.3704 & 0.8828 & 0.2382 & 0.2807 & 0.5057 & -0.0854 & 0.5360 & -0.1602 \\ 0.9610 & 0.0977 & 0.5585 & 0.0080 & 0.3308 & 0.3902 & 0.1012 & 0.1440 & 0.3834 & 0.8618 \\ 0.0281 & 0.4562 & 0.3044 & 0.2126 & 0.6124 & 0.5310 & 0.6431 & 0.8487 & 0.3293 & 0.1125 \\ 0.0108 & 0.4460 & 0.1372 & 0.7794 & 0.0567 & 0.0788 & 0.2557 & 0.0073 & 0.2873 & 0.0257 \\ 0.1644 & 0.2111 & -0.4123 & -0.0413 & 0.4501 & 0.4552 & -0.2551 & -0.3496 & 0.3553 & 0.3114 \\ -0.1021 & 0.2088 & 0.2768 & -0.0792 & 0.1370 & 0.1753 & 0.1609 & 0.0324 & 0.3319 & 0.1487 \\ -0.0175 & 0.4511 & -0.2043 & 0.4071 & 0.1864 & 0.2045 & -0.4055 & -0.0786 & 0.3076 & 0.0537 \end{bmatrix}$$

SCDL O(540)

$$\widehat{\mathbf{X}} = \begin{bmatrix} -0.0506 & 0.5894 & -0.9616 & 0.5009 & 0.7861 & -0.3790 \\ 0.4828 & 0.7520 & -0.2699 & -0.7523 & 0.3104 & 0.9214 \\ 0.8743 & 0.2952 & -0.0491 & 0.4280 & 0.5345 & -0.0857 \\ 0.0026 & 0.3474 & 0.9247 & 0.2509 & 0.6179 & 0.1436 \\ 0.2331 & 0.5654 & 0.0729 & 0.5660 & 0.0964 & 0.8490 \\ 0.7644 & 0.0871 & 0.0024 & 0.1831 & 0.2857 & 0.0073 \\ -0.0244 & 0.4432 & 0.2596 & -0.3768 & 0.2440 & -0.3492 \\ -0.0443 & 0.1740 & 0.0472 & 0.2144 & 0.4202 & 0.0325 \\ 0.4221 & 0.2219 & 0.0133 & -0.3220 & 0.1659 & -0.0789 \end{bmatrix}$$

$\longrightarrow \widehat{\mathbf{X}} = \begin{bmatrix} -0.0506 & 0.5894 & -0.9616 & 0.5009 & 0.7861 & -0.3790 \\ 0.4828 & 0.7520 & -0.2699 & -0.7523 & 0.3104 & 0.9214 \\ 0.8743 & 0.2952 & -0.0491 & 0.4280 & 0.5345 & -0.0857 \end{bmatrix}$

$$\widetilde{\mathbf{W}} = \begin{bmatrix} 0.0618 & 0.4899 & 0 & 0.9497 & 0 & 0 & 0 & 0.1078 & 0 \\ 0 & 0.4723 & 0 & 0 & 0.9402 & 0.9384 & 0 & 0.4296 & 0 \\ 0.8991 & 0 & 0 & 0 & 0 & 0 & 0.0511 & 0 & 0.9478 \\ 0 & 0 & 0.7650 & 0 & 0 & 0 & 0.9078 & 0 & 0 \\ 0 & 0.0081 & 0.2012 & 0 & 0 & 0.0124 & 0 & 0.4267 & 0 \\ 0 & 0 & 0 & 0 & 0.0105 & 0 & 0 & 0.9500 & 0 \end{bmatrix}$$

NNZE(W̃) = 19 < NNZE(Y)

$AE_{original} = \frac{1}{2}\|\mathbf{Y} - \mathbf{XW}\|_F^2 < 0.1$, yields the dictionary matrix $\mathbf{X}$ and the sparse matrix $\mathbf{W}$. In this case, $\mathbf{W}$ includes 40 nonzero elements (i.e., NNZE($\mathbf{W}$)=40), which exceeds the amount of data in the original $\mathbf{Y}$ (NNZE($\mathbf{Y}$)=30). Therefore, $\mathbf{W}$ cannot be considered as a sparse form of $\mathbf{Y}$.

**Elevating Data Dimensions:** Inspired by the example in Fig. 2, elevating the dimensionality of data may potentially enhance the effectiveness of sparse coding. In Example 1, we directly extend each $\mathbf{y}_i = [y_{1i}, y_{2i}, y_{3i}]^T$ vertically to $\widetilde{\mathbf{y}}_i = [y_{1i}, y_{2i}, y_{3i}, y_{1i}^2, y_{2i}^2, y_{3i}^2, y_{1i}y_{2i}, y_{1i}y_{3i}, y_{2i}y_{3i}]^T$, for $i = 1, 2, \cdots, 10$, forming a high-dimensional data matrix $\widetilde{\mathbf{Y}}$. Given that the dictionary matrix is composed of 6 atoms, conducting sparse coding and dictionary learning on $\widetilde{\mathbf{Y}}$ in the high-dimensional feature space, with the constraint that the approximation error $AE_{original} = \frac{1}{2}\|\mathbf{Y} - \widehat{\mathbf{X}}\widetilde{\mathbf{W}}\|_F^2 < 0.1$, yields the dictionary matrix $\widetilde{\mathbf{X}}$ and the sparse matrix $\widetilde{\mathbf{W}}$. Note that, just as $\widetilde{\mathbf{Y}}$ is a vertical expansion of $\mathbf{Y}$, $\widetilde{\mathbf{X}}$ is a vertical expansion of $\widehat{\mathbf{X}}$. In this case, there are only 19 nonzero elements in $\widetilde{\mathbf{W}}$ (i.e., NNZE($\widetilde{\mathbf{W}}$)=19), which is less than that of the original data $\mathbf{Y}$ (NNZE($\mathbf{Y}$)=30). Consequently, $\widetilde{\mathbf{W}}$ can be regarded as a sparse representation of $\mathbf{Y}$. That is, **expanding dimensions can indeed result in a more efficient sparse representation of the original data**.

Indeed, there are numerous nonlinear mapping methods that can be utilized to enhance the dimensionality of data. For instance, similar to the approach in Example 1 where all second-order terms of the original data are introduced, we could also incorporate all third-order terms, all fourth-order terms, and so on. Additionally, we could introduce a combination of all second-order and third-order terms, or selectively introduce parts of second-order and third-order terms, and so forth. In essence, the ways to increase dimensionality are virtually limitless, and it is impractical to verify each method individually to determine which best meets practical needs. Therefore, **the specific form and scale of data dimensionality enhancement that can enhance the efficiency of sparse coding remains a thought-provoking issue**.

Theoretically, the more nonlinear terms that are incorporated, the more likely it is to enhance the efficiency of sparse coding. However, **as the dimensionality of the data increases, the computational complexity inevitably rises as well**. As shown in Example 1, the computational complexity of sparsely coding $\mathbf{Y}$ into $\mathbf{X}$ and $\mathbf{W}$ is $O(180)$. After expanding $\mathbf{Y}$ into $\widetilde{\mathbf{Y}}$, the computational complexity of sparsely coding $\widetilde{\mathbf{Y}}$ into $\widetilde{\mathbf{X}}$ and $\widetilde{\mathbf{W}}$ increases to $O(540)$. If the scale of $\mathbf{Y}$ is too large or if the data expansion method is too complex, the corresponding computational load may become prohibitive. Therefore, overcoming the computational complexity introduced by data dimensionality expansion remains a significant challenge.

**Advantages of RBF Kernel in Elevating Data Dimensions:** In response to the aforementioned challenges of "indeterminate forms and scales, and high computational complexity of data dimensionality enhancement", in this section, we validate that the Radial Basis Function (RBF) kernel possesses excellent properties that give it distinct advantages in elevating data dimensions.

For any $\mathbf{a}, \mathbf{b} \in R^n$, their RBF kernel $\kappa(\mathbf{a}, \mathbf{b}) = \exp(-\frac{1}{2\sigma^2}\|\mathbf{a} - \mathbf{b}\|_2^2)$ can be decomposed as

$$\kappa(\mathbf{a}, \mathbf{b}) = \exp\left(-\frac{1}{2\sigma^2}\left(\|\mathbf{a}\|_2^2 + \|\mathbf{b}\|_2^2\right)\right)\exp\left(\frac{\mathbf{a}^T\mathbf{b}}{\sigma^2}\right) = \phi(\mathbf{a})^T\phi(\mathbf{b}) \qquad (2)$$

where $\sigma$ is the RBF kernel hyper-parameter, $\phi(\mathbf{z}) = \left[\sqrt{\frac{\varrho}{\sigma^{2q}q_1!\cdots q_n!}}\left(z_1^{q_1}\cdots z_n^{q_n}\right)\right]_{|\mathbf{q}|=0}^{\infty}$ is an infinite order polynomial map for $\mathbf{z} = \mathbf{a}$ or $\mathbf{b}$, $|\mathbf{q}| = q_1 + \cdots + q_n$, and $\varrho = \exp\left(-\frac{1}{2\sigma^2}\left(\|\mathbf{a}\|_2^2 + \|\mathbf{b}\|_2^2\right)\right)$.

Notably, on the one hand, $\phi(\mathbf{z})$ includes all integer-order terms of the original data $\mathbf{z}$, including infinite order. That is to say, $\phi(\mathbf{z})$ **contains all forms of data expansion, and the scale of the expanded data is infinite**. On the other hand, the inner product of two expanded data points $\phi(\mathbf{a})$ and $\phi(\mathbf{b})$ can be directly calculated through kernel function $\kappa(\mathbf{a}, \mathbf{b})$ with computation complexity of $O(n)$, instead of $\phi(\mathbf{a})^T \phi(\mathbf{b})$ with computation complexity of $\infty$. That is to say, in the calculation process, **as long as the inner product of extended data points is formed, the problem of computational complexity resulting from dimensionality increase can be avoided via kernel tricks**. Precisely because of these great properties, the RBF kernel is expected to exhibit excellent performance in expanding data dimensions. Consequently, our endeavor in this paper revolves around leveraging the feature mapping $\phi$, inherent to the RBF kernel, to expand data dimensions, ultimately enhancing the performance of sparse coding.

## 3 KERNEL SPARSE CODING DICTIONARY LEARNING

Based on the discussion above, in order to achieve a more efficient sparse coding scheme for the input data $\mathbf{Y}$, we impose the feature mapping in (2) to the historical data $\mathbf{Y}$ and the dictionary matrix $\mathbf{X}$, formulating the following Kernel Sparse Coding Dictionary Learning (KSCDL) problem:

$$\min_{\mathbf{X},\mathbf{W}} \frac{1}{2}\|\phi(\mathbf{Y}) - \phi(\mathbf{X})\mathbf{W}\|_F^2 + \lambda \sum_{j=1}^{m} \|\mathbf{w}_j\|_1. \tag{3}$$

It is noteworthy that, although the KSCDL in (3) is an extension of (1), the solution of (3) can not be obtained by extending the solution of (1). For an in-depth discussion, please refer to A.3.

**Solution Overview:** To solve KSCDL (3), we decompose the KSCDL problem into a Kernel Sparse Coding (KSC) subproblem and a Kernel Dictionary Learning (KDL) subproblem:
• **KSC:** The KSC subproblem of (3) can be interpreted as finding the weight matrix $\mathbf{W}$, while keeping the dictionary matrix $\mathbf{X}$ fixed, through

$$\mathbf{W} = \arg\min_{\mathbf{W}} \frac{1}{2} \sum_{j=1}^{m} \left[ \|\phi(\mathbf{y}_j) - \phi(\mathbf{X})\mathbf{w}_j\|_2^2 + \lambda\|\mathbf{w}_j\|_1 \right]. \tag{4}$$

• **KDL:** The KDL subproblem of (3) involves solving the dictionary matrix $\mathbf{X}$, with the sparse coding matrix $\mathbf{W}$ fixed, through

$$\mathbf{X} = \arg\min_{\mathbf{X}} \frac{1}{2}\|\phi(\mathbf{Y}) - \phi(\mathbf{X})\mathbf{W}\|_F^2. \tag{5}$$

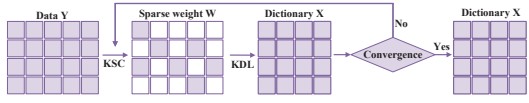

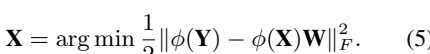

Figure 3: Flowchart of KSCDL.

Moving forward, based on the block coordinate descent method, we alternately and iteratively solve these two sub-problems as shown in Fig. 3. Upon convergence of the algorithm, we output and save the dictionary matrix $\mathbf{X}$.

**Solution for KSC:** For each column of $\mathbf{W}$ and $\mathbf{Y}$ (abbreviated as $\mathbf{w}$ and $\mathbf{y}$, respectively), the objective function in (4) can be equivalent rewritten as:

$$\min_{\mathbf{w}} \frac{1}{2}\|\phi(\mathbf{y}) - \phi(\mathbf{X})\mathbf{w}\|_2^2 + \lambda\|\mathbf{w}\|_1, \tag{6}$$

Based on the kernel trick, we have $\|\phi(\mathbf{y}) - \phi(\mathbf{X})\mathbf{w}\|_2^2 = \kappa_{\mathbf{yy}} - 2\kappa_{\mathbf{yX}}\mathbf{w} + \mathbf{w}^T\kappa_{\mathbf{XX}}\mathbf{w}$. In which $\kappa_{\mathbf{yy}} = \kappa(\mathbf{y}, \mathbf{y}) = \phi(\mathbf{y})^T\phi(\mathbf{y})$, $\kappa_{\mathbf{yX}} = \kappa(\mathbf{y}, \mathbf{X}) = \phi(\mathbf{y})^T\phi(\mathbf{X}) = [\kappa(\mathbf{y}, \mathbf{x}_1), \cdots, \kappa(\mathbf{y}, \mathbf{x}_p)]$, and $\kappa_{\mathbf{XX}} = \phi(\mathbf{X})^T\phi(\mathbf{X})$ with $[\kappa(\mathbf{X}, \mathbf{X})]_{ij} = \kappa(\mathbf{x}_i, \mathbf{x}_j)$. Then problem (6) is equivalent to

$$\min_{\mathbf{w}} -\kappa_{\mathbf{yX}}\mathbf{w} + \frac{1}{2}\mathbf{w}^T\kappa_{\mathbf{XX}}\mathbf{w} + \lambda\|\mathbf{w}\|_1. \tag{7}$$

Denote the objective function of problem (7) as $\mathfrak{J}$, if $w_i > 0$, the derivative of $\mathfrak{J}$ with respect to $w_i$ can be obtained as $\frac{\partial \mathfrak{J}}{\partial w_i} = -\kappa_{\mathbf{yx}_i} + \sum_{j=1}^{p} \kappa_{ij} w_j + \lambda$, where $\kappa_{\mathbf{yx}_i} = \kappa(\mathbf{y}, \mathbf{x}_i)$ and $\kappa_{ij} = \kappa(\mathbf{x}_i, \mathbf{x}_j)$. Let $\frac{\partial \mathfrak{J}}{\partial w_i} = 0$, one gets $-\kappa_{\mathbf{yx}_i} + \sum_{j\neq i}^{p} \kappa_{ij} w_j + \kappa_{ii} w_i + \lambda = 0$. That is $w_i = \kappa_{\mathbf{yx}_i} - \sum_{j\neq i}^{p} \kappa_{ij} w_j - \lambda$, since $\kappa_{ii} = 1$. For convenience, we denote $z_i = \kappa_{\mathbf{yx}_i} - \sum_{j\neq i}^{p} \kappa_{ij} w_j$. To avoid confusion, we refer to one complete iteration of the coordinate descent algorithm as a round, which starts from updating

$w_1$ and continues until $w_p$ has been updated. The variables updated in the $k$th round will be marked with a superscript $(k)$, then

$$z_i^{(k)} = \kappa_{\mathbf{yx}_i} - \sum_{j \neq i}^{p} \kappa_{ij} w_j^{[i,k]}, \text{ where } w_j^{[i,k]} = \begin{cases} w_j^{(k)}, & \text{if } j < i; \\ w_j^{(k-1)}, & \text{if } j > i. \end{cases} \quad (8)$$

By analogy, it's not difficult to summarize that

$$w_i^{(k)} = \text{soft}_\lambda(z_i^{(k)}) = \begin{cases} z_i^{(k)} - \lambda, & \text{if } z_i^{(k)} > \lambda; \\ z_i^{(k)} + \lambda, & \text{if } z_i^{(k)} < -\lambda; \\ 0, & \text{if } |z_i^{(k)}| \leq \lambda. \end{cases} \quad (9)$$

**Solution for KDL:** Based on the kernel trick, the objective function in (5) can be rewritten as

$$\|\phi(\mathbf{Y}) - \phi(\mathbf{X})\mathbf{W}^{(k)}\|_F^2 = \text{Trace}\Big[\kappa_{\mathbf{YY}} - 2\kappa_{\mathbf{YX}}\mathbf{W}^{(k)} + (\mathbf{W}^{(k)})^T \kappa_{\mathbf{XX}}\mathbf{W}^{(k)}\Big]. \quad (10)$$

where $\kappa_{\mathbf{YY}} = \kappa(\mathbf{Y}, \mathbf{Y}) = \phi(\mathbf{Y})^T\phi(\mathbf{Y})$, $\kappa_{\mathbf{YX}} = \kappa(\mathbf{Y}, \mathbf{X}) = \phi(\mathbf{Y})^T\phi(\mathbf{X})$. There is no closed form solution for dictionary matrix $\mathbf{X}$ due to the kernel matrices, we update $\mathbf{X}$ via the gradient descent method. The gradient of the objective function w.r.t. matrix $\mathbf{X}$ can be calculated as $G_{\mathbf{X}} = \frac{1}{\sigma^2}(\mathbf{X}\Gamma_1 - \mathbf{Y}\mathbf{Q}_1 + \mathbf{X}\mathbf{Q}_2 - \mathbf{X}\Gamma_2)$. Here $\mathbf{Q}_1 = (\mathbf{W}^{(k)})^T \odot \kappa_{\mathbf{YX}}$, $\Gamma_1 = \text{diag}(\mathbf{1}_n^T\mathbf{Q}_1)$, $\mathbf{Q}_2 = \mathbf{W}^{(k)}(\mathbf{W}^{(k)})^T \odot \kappa_{\mathbf{XX}}$, $\Gamma_2 = \text{diag}(\mathbf{1}_r^T\mathbf{Q}_2)$. We update $\mathbf{X}$ as

$$\mathbf{X}^{(k)} \leftarrow \mathbf{X}^{(k-1)} - \mu G_{\mathbf{X}}, \quad (11)$$

where $\mu$ is the optimization stepsize. As outlined in Fig.3, the organization details of the KSCDL algorithm and the theoretical guarantee of its convergence are provided in A.2.

# 4 FAST KERNEL SPARSE CODING

The KSC subproblem is solved basing on the coordinate descent principle, which guarantees convergence. However, the updating iterative process of the weight vector is slow.

**Computational complexity analysis.** In Fig. 4, we provide a illustration of the update process for sparse weight vector $\mathbf{w}$ during the $k$th round of iteration. It can be observed that the calculation of the intermediate variable $z_i^{(k)} = \kappa_{\mathbf{yx}_i} - \sum_{j \neq i}^{p} \kappa_{ij} w_j^{[i,k]}$, for each sparse weight factor $w_i^{(k)}$, has a complexity of $O(p)$, and there are $p$ such weight factors (i.e., $w_i^{(k)}, i = 1, 2, \cdots, p$) in each weight vector $\mathbf{w}$. Therefore, the computational complexity for each iteration of $\mathbf{w}$ is $O(p^2)$, making the update process relatively sluggish. In the following, we will approach from two aspects: theoretical analysis and experimental observation, to accelerate the KSC process in two stages.

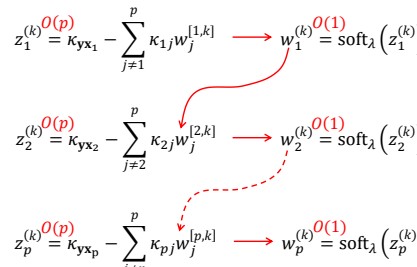

Figure 4: Standard updating process for sparse weight vector $\mathbf{w}$.

## 4.1 ACCELERATION STRATEGY I: SAFELY SKIPPING UPDATES FOR ZERO ELEMENTS

**Theoretical Analysis:** Equation (9) shows that $w_i^{(k)}$ is essentially a soft threshold function of $z_i^{(k)}$ with the threshold $\lambda$. It is particularly important to note that if $|z_i^{(k)}| \leq \lambda$ holds, we can directly obtain $w_i^{(k)} = 0$. In other words, there may be a scenario in which $w_i^{(k)}$ does not require an update calculation and can be set to 0 directly. In order to anticipate the relationship between $z_i^{(k)}$ and $\lambda$, we intend to explore the upper and lower bounds of $z_i^{(k)}$. For this purpose, we recompute $z_i^{(k)}$ as:

$$z_i^{(k)} = z_i^{(k-1)} - \kappa_{i:}\Delta\mathbf{w}_{(i)}^{(k)}, \quad (12)$$

where $\kappa_{i:} = [\kappa_{i1}, \kappa_{i2}, \cdots, \kappa_{ip}]$, $\Delta\mathbf{w}_{(i)}^{(k)} = [\Delta w_1^{(k)}, \cdots, \Delta w_{i-1}^{(k)}, 0, \Delta w_{i+1}^{(k-1)}, \cdots, \Delta w_p^{(k-1)}]^T$ and $\Delta w_l^{(k)} = w_l^{(k)} - w_l^{(k-1)}$ for each $l = 1, \cdots, p$. Combining with the Cauchy-Schwarz inequality (Steele, 2004), it can be obtained from (12) that

$$\underline{z}_i^{(k)} \leq z_i^{(k)} \leq \overline{z}_i^{(k)} \quad (13)$$

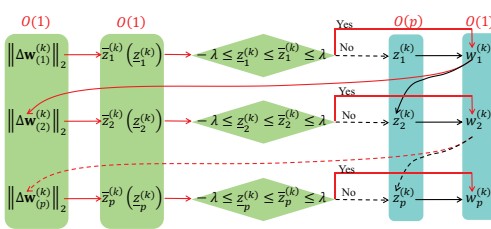

Figure 5: Acceleration Strategy I.



Figure 6: Complete update process of $\mathbf{w}$ based on Acceleration Strategy I.

where

$$\overline{z}_i^{(k)} = z_i^{(k-1)} + \|\kappa_{i:}\|_2 \|\Delta\mathbf{w}_{(i)}^{(k)}\|_2, \text{ and } \underline{z}_i^{(k)} = z_i^{(k-1)} - \|\kappa_{i:}\|_2 \|\Delta\mathbf{w}_{(i)}^{(k)}\|_2. \tag{14}$$

Obviously, based on $\overline{z}_i^{(k)}$, the computational complexity of $\underline{z}_i^{(k)}$ is $O(1)$. Naturally, we have

• If $\underline{z}_i^{(k)} > \lambda$ or $\overline{z}_i^{(k)} < -\lambda$, then $w_i^{(k)} \neq 0$, meaning $w_i$ needs to undergo an update calculation.

• If $-\lambda \leq \underline{z}_i^{(k)} \leq \overline{z}_i^{(k)} \leq \lambda$, then $w_i^{(k)} = 0$, indicating that the update for $w_i$ can be avoided.

The second scenario mentioned above indicates that there are indeed situations where it is not necessary to compute $z_i^{(k)}$, and only the calculations of $\overline{z}_i^{(k)}$ and $\underline{z}_i^{(k)}$ are needed to determine that $w_i^{(k)} = 0$, which can effectively reduce computational complexity.

In fact, $z_i^{(k-1)}$ can be directly obtained from the $(i-1)$th round and $\|\kappa_{i:}\|_2$ can be precomputed before iterations. In addition, based on the definition of $\Delta\mathbf{w}_{(i)}^{(k)}$ in equation (12), we can get

$$\|\Delta\mathbf{w}_{(i)}^{(k)}\|_2 = \sqrt{\|\Delta\mathbf{w}_{(i-1)}^{(k)}\|_2^2 - (\Delta w_i^{(k-1)})^2 + (\Delta w_{i-1}^{(k)})^2}. \tag{15}$$

That is, once we obtain $\|\Delta\mathbf{w}_{(i-1)}^{(k)}\|_2$, the computational complexity of $\|\Delta\mathbf{w}_{(i)}^{(k)}\|_2$ can be ignored. Therefore, for the $k$th round, the total computational complexity of $\overline{z}_i^{(k)}$ and $\underline{z}_i^{(k)}$, for all $i \in \{1, \cdots, p\}$, primarily arises from $\|\Delta\mathbf{w}_{(1)}^{(k)}\|_2^2$. Further more, we have

$$\|\Delta\mathbf{w}_{(1)}^{(k)}\|_2 = \sqrt{\|\Delta\mathbf{w}_{(p)}^{(k-1)}\|_2^2 - (\Delta w_1^{(k-1)})^2 + (\Delta w_p^{(k-1)})^2}. \tag{16}$$

That is, once we obtain $\|\Delta\mathbf{w}_{(p)}^{(k-1)}\|_2$, the computational complexity of $\|\Delta\mathbf{w}_{(1)}^{(k)}\|_2$ can be ignored. By analogy, for any $i \in \{1, \cdots, p\}$ and integer $k \geq 1$, the computational complexity of $\|\Delta\mathbf{w}_{(i)}^{(k)}\|_2$ primarily stems from $\|\Delta\mathbf{w}_{(1)}^{(1)}\|_2$, which is determined by the initialization rule of the algorithm. It can be 0 if we initialize $\Delta\mathbf{w}$ as a zero vector, or it can be $O(p)$ if we initialize $\Delta\mathbf{w}$ as a nonzero vector. From this perspective, using the upper and lower bounds $\overline{z}_i^{(k)}$ and $\underline{z}_i^{(k)}$ to anticipate zero elements in $\mathbf{w}$ holds the promise of reducing computational complexity.

**Acceleration Strategy Design:** Based on the above theoretical analysis, we propose to utilize $\overline{z}_i^{(k)}$ and $\underline{z}_i^{(k)}$ to accelerate the update of zero elements in the weight vector $\mathbf{w}$. As shown in Fig. 5, before calculating the intermediate variable $z_i^{(k)}$ for $w_i^{(k)}$, we first calculate the the upper and lower bounds of $z_i^{(k)}$ and use them to predict whether $w_i^{(k)}$ is zero. If it is, we directly skip the update of $z_i^{(k)}$; if not, we calculate $z_i^{(k)}$ and thereby update $w_i^{(k)}$. It is worth noting that **this acceleration strategy is not restricted to the case using RBF kernels, as w is calculated from $\kappa_{\mathbf{yX}}$ and $\kappa_{\mathbf{XX}}$, which are not constrained by any specific kernel function**.

### 4.2 ACCELERATION STRATEGY II: PRIORITIZING UPDATES FOR INITIALLY PREDICTED NONZERO ELEMENTS

**Experimental Observation:** Based on Acceleration Strategy I, if we update the elements of the weight vector $\mathbf{w}$ following the process shown in Fig. 6, and track the positions of nonzero elements

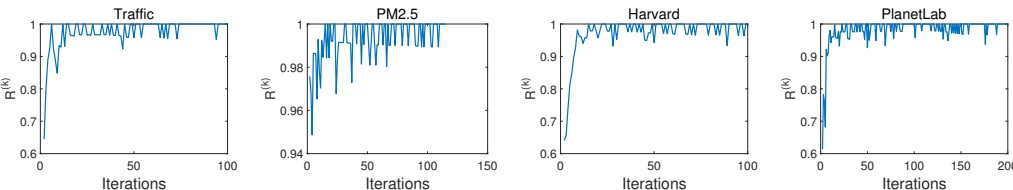

Figure 7: Changes for $R^{(k)}$ during the iteration process.

in $\mathbf{w}$, we will find that the positions of nonzero elements appear to be relatively stable. In other words, **the nonzero elements in vector $\mathbf{w}^{(k+1)}$ occupy positions that are closely analogous to those in vector $\mathbf{w}^{(k)}$.** To better describe this phenomenon, we define several new metrics to track the changes in the positions of nonzero elements throughout the iterative process:

- $\mathbb{I}^{(k)}$: the index set of nonzero elements in $\mathbf{w}^{(k)}$.
- $|\mathbb{I}^{(k)}|$: the number of nonzero elements in $\mathbf{w}^{(k)}$.
- $\mathbb{I}^{(k)} \cap \mathbb{I}^{(k+1)}$: the index set of elements that are nonzero in $\mathbf{w}^{(k)}$ and also nonzero in $\mathbf{w}^{(k+1)}$.
- $R^{(k)} = |\mathbb{I}^{(k)} \cap \mathbb{I}^{(k+1)}|/|\mathbb{I}^{(k)}|$: the ratio of the number of elements that are nonzero in $\mathbf{w}^{(k)}$ and remain nonzero in $\mathbf{w}^{(k+1)}$, to the number of nonzero elements in $\mathbf{w}^{(k)}$.

Fig. 7 illustrates the variation process of the metric $R^{(k)}$ on four real-word datasets, from which we can see that, in each iteration of the loop, over 60% of the nonzero elements maintain their positions. We call this feature **relative stability** of nonzero element positions.

Given that the positions of nonzero elements exhibit **relative stability**, this indicates that in the process of adjacent continuous multiple rounds of iterations, only a subset of nonzero elements with fixed positions are being updated. For example, as shown in Fig. 6, in the first round of iteration, only $w_i$ and $w_j$ are predicted as nonzero elements, while all other elements are zero. And in the first $k$ rounds of iteration, only $w_i$ and $w_j$ necessitate updates. This situation renders the prediction of the remaining zero elements somewhat unnecessary.

Although the computational complexity of prediction behavior is not high, the more rounds required to update these nonzero elements, the more computing resources are wasted, especially for large-scale networks.

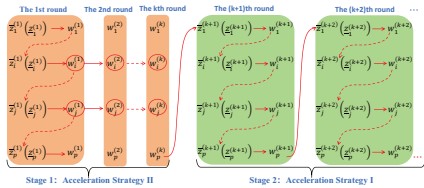

**Acceleration Strategy Design:** To alleviate the situation above, we proposed to focus on updating the nonzero weights filtered out by $\overline{z}_i^{(k)}$ and $\underline{z}_i^{(k)}$ until convergence initially, as shown in Stage 1 in Fig. 8. The complete organization of FKSC algorithm and its computational complexity analysis are detailed in A.4.

Figure 8: Overview of the two-stage acceleration strategy.

## 5 EXPERIMENT

Overall, our experimentation encompasses an initial Kernel Sparse Coding Dictionary Learning (KSCDL) training phase and a subsequent Fast Kernel Sparse Coding (FKSC) testing phase. The specific details of the experimental setup, including the dataset, comparison algorithms, and evaluation metrics, can be found in A.5. Within the initial KSCDL training phase, we replace KSC with FKSC, verifying the convergence of the KSCDL method and analyzing the impact of various hyperparameters on the KSCDL objective value in A.6. Moving on to the subsequent FKSC testing phase, we proceed to validate the effectiveness of the acceleration strategy within FKSC, and compare its performance in the context of data compression (the specific implementation steps are outlined in A.7) with other existing algorithms. The corresponding results are shown in the following.

**Effectiveness of Acceleration Strategy:** The effectiveness of the acceleration strategy is manifested in two aspects: accuracy and processing time. In the following, we verify that the proposed FKSC method can accelerate processing time while ensuring there is no additional loss of accuracy.

Given the limited scale of real datasets, which may hinder the effectiveness of our acceleration strategy, thus, we integrate synthetic data, gradually increasing the data scale, to verify the scalability performance of the proposed algorithm. Specifically, we generate synthetic data, denoted as Syn600, Syn800, Syn1000, Syn1200 and Syn1400 respectively, following the generation method of Syn360

in A.5. The data sizes used for training in them are $600 \times 1200, 800 \times 1600, 1000 \times 2000, 1200 \times 2400, 1400 \times 2800$, and the data sizes for testing are $600 \times 600, 800 \times 800, 1000 \times 1000, 1200 \times 1200, 1400 \times 1400$, with the dictionary matrix configured as a square and hyperparameters set to $\lambda = 0.01$ and $\sigma = 20$ uniformly.

The experimental results are shown in Table 1 and Fig. 9, from which we can observe that

• The two-stage accelerated strategy in FKSC does not exert significantly detrimental effect on accuracy. As shown in Table 1, the objective values remain consistent between FKSC and KSC across the evaluated datasets.

• The FKSC method can significantly reduce processing time. Fig. 9 (a) shows that FKSC has effectively improved the efficiency of KSC, achieving a remarkable reduction in processing time by up to 90%, for each

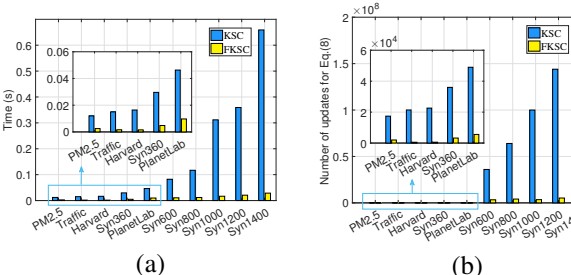

(a)                    (b)

Figure 9: FKSC's scalability experiments. For each column in sparse weight matrix $\mathbf{W}$: (a) the average processing time; (b) the average number of updates for (8).

column in sparse weight matrix $\mathbf{W}$. This improvement is attributed to FKSC's strategy of calculating the upper and lower bounds of intermediate variables, thereby substantially circumventing their updates, as illustrated in Fig. 9 (b).

• As the data size increases, the acceleration effect of FKSC becomes increasingly evident. As shown in Fig. 9 (a), the processing time for both KSC and FKSC increases in tandem with the expansion of data scale. Notably, the KSC method experiences a swift surge in processing time, contrasted by a more gradual increment for the FKSC method. As a result, the gap in processing time between KSC and FKSC becomes increasingly pronounced as the data scale increases.

Table 1: The objective values for KSC and FKSC

|  | PM2.5 | Traffic | Harvard | Syn360 | PlanetLab | Syn600 | Syn800 | Syn1000 | Syn1200 | Syn1400 |
|---|---|---|---|---|---|---|---|---|---|---|
| KSC | 0.2761 | 0.1378 | 0.3877 | 1.8439 | 0.8328 | 1.6057 | 2.1432 | 2.6820 | 3.2092 | 3.7574 |
| FKSC | 0.2761 | 0.1378 | 0.3877 | 1.8439 | 0.8328 | 1.6057 | 2.1432 | 2.6820 | 3.2092 | 3.7574 |

**Compression Performance:** Given that FKSC effectively converts dense data into sparse data comprising merely a few non-zero elements, we delve into its application within the realm of data compression. The intricate implementation details are comprehensively presented in A.7.

The Compression Ratio (CR) is determined by FKSC, which hinges critically on the sparse penalty parameter $\lambda$. Specifically, as stated in (9), the larger the value of $\lambda$, the greater the interval $[-\lambda, \lambda]$ becomes. Consequently, more intermediate variables $z_i(i = 1, 2, \cdots, p)$ fall within this interval, leading to a higher number of zero weights. As a result, $\mathbf{w}$ becomes sparser, and the compression ratio increases. However, there is a trade-off as an excessively large $\lambda$ can render $\mathbf{w}$ overly sparse, potentially compromising the accuracy of data reconstruction.

Motivated by the experiments detailed in A.6, we take $\sigma = 10, 4, 11, 17$ and 10 for PM2.5, Traffic, Harvard, Syn360 and PlanetLab respectively, and confine the penalty parameter $\lambda$ within the interval $[0.001, 0.1]$ to ensure that the sparse weight matrix $\mathbf{W}$ is not too sparse to recover the original data $\mathbf{Y}$. In Table 2, we detail the compression performance parameters of various algorithms across five datasets at four distinct compression ratios (CRs), with the optimal and sub-optimal performance parameters highlighted in bold. From which, several key observations emerge:

• Superiority of FKSC over CD: With the exception of the PM2.5 dataset at $\mathrm{CR} = 26.0293$, the FKSC method consistently outperforms CD in all three performance metrics. Notably, in the Traffic data at $\mathrm{CR} = 57.3321$, i.e., the data size is compressed to 1.75% of the original scale, the NMAE, NMAE, NECR metrics of FKSC reach 0.1764, 0.1259, 6057 respectively, achieving a remarkable 21% relative improvement in NRMSE, 47% in NMAE, and an astonishing three times increase in NECR. This underscores the significant enhancement in sparse coding performance brought about by the incorporation of kernel trick.

• Competitive performance of FKSC compared to KSC: It is evident that these restoration performance parameters of FKSC closely align with those of KSC across five datasets at four different CRs. This indicates that the two-stage acceleration strategy in FKSC exhibits virtually no adverse impact on data compression.

Table 2: Compression performance of FKSC and other comparison methods.

**Traffic**

| Traffic | CR=30.5866 | | | CR=40.3351 | | | CR=50.1481 | | | CR=57.3321 | | |
|---|---|---|---|---|---|---|---|---|---|---|---|---|
| | NRMSE | NMAE | NECR | NRMSE | NMAE | NECR | NRMSE | NMAE | NECR | NRMSE | NMAE | NECR |
| CD | 0.1902 | 0.1569 | 2967 | 0.2012 | 0.1708 | 2285 | 0.2077 | 0.1781 | 2238 | 0.2135 | 0.1857 | 1900 |
| FKSC | **0.1723** | **0.1242** | **6077** | **0.1742** | **0.1250** | 5980 | **0.1760** | **0.1251** | 6184 | **0.1764** | **0.1259** | **6057** |
| KSC | **0.1723** | **0.1242** | 6075 | **0.1742** | **0.1250** | 5982 | **0.1760** | **0.1251** | 6183 | **0.1764** | **0.1259** | **6057** |
| KFMC | 0.2026 | 0.1611 | 4007 | 0.2018 | 0.1596 | 4240 | 0.1986 | 0.1525 | 4600 | 0.2092 | 0.1619 | 4189 |
| KSR-$L_{21}$ | 0.1760 | 0.1290 | 5989 | 0.1857 | 0.1339 | **6233** | 0.2048 | 0.1444 | 5476 | 0.2476 | 0.1761 | 4402 |
| LPM | 0.2799 | 0.2323 | 2589 | 0.2815 | 0.2342 | 2430 | 0.2709 | 0.2253 | 2534 | 0.2797 | 0.2320 | 2665 |

**PM2.5**

| PM2.5 | CR=26.0293 | | | CR=32.1060 | | | CR=41.9058 | | | CR=51.1438 | | |
|---|---|---|---|---|---|---|---|---|---|---|---|---|
| | NRMSE | NMAE | NECR | NRMSE | NMAE | NECR | NRMSE | NMAE | NECR | NRMSE | NMAE | NECR |
| CD | 0.2581 | 0.2078 | **2168** | 0.2877 | 0.2373 | 1689 | 0.3075 | 0.2565 | 1377 | 0.3282 | 0.2769 | 1303 |
| FKSC | **0.2519** | **0.2060** | 2122 | **0.2575** | **0.2122** | **2025** | **0.2625** | **0.2153** | 2077 | **0.2665** | **0.2195** | **2057** |
| KSC | **0.2519** | **0.2060** | 2122 | **0.2575** | **0.2122** | 2023 | **0.2625** | **0.2153** | 2078 | **0.2665** | **0.2195** | **2057** |
| KFMC | 0.2643 | 0.2248 | 1601 | 0.2656 | 0.2285 | 1567 | 0.2861 | 0.2447 | 1612 | 0.2711 | 0.2304 | 1728 |
| KSR-$L_{21}$ | 0.2803 | 0.2358 | 1576 | 0.4054 | 0.3152 | 1249 | 0.3110 | 0.2634 | 1523 | 0.3539 | 0.2867 | 1426 |
| LPM | 0.3781 | 0.3421 | 910 | 0.3471 | 0.3127 | 1044 | 0.3308 | 0.2923 | 1255 | 0.3135 | 0.2759 | 1352 |

**Harvard**

| Harvard | CR=40.3763 | | | CR=54.1633 | | | CR=63.2912 | | | CR=74.4482 | | |
|---|---|---|---|---|---|---|---|---|---|---|---|---|
| | NRMSE | NMAE | NECR | NRMSE | NMAE | NECR | NRMSE | NMAE | NECR | NRMSE | NMAE | NECR |
| CD | 0.2083 | 0.1774 | 3924 | 0.2181 | 0.1915 | 3187 | 0.2365 | 0.2173 | 2884 | 0.2435 | 0.2261 | 2737 |
| FKSC | **0.2028** | **0.1654** | **4971** | **0.2057** | **0.1675** | **4481** | **0.2106** | **0.1760** | 3868 | **0.2130** | **0.1786** | **3836** |
| KSC | **0.2028** | **0.1654** | **4971** | **0.2057** | **0.1675** | **4481** | **0.2106** | **0.1760** | 3868 | **0.2130** | **0.1786** | **3836** |
| KFMC | 0.3119 | 0.2163 | 2907 | 0.2686 | 0.2282 | 3298 | 0.3536 | 0.3017 | 3137 | 0.4250 | 0.3129 | 3215 |
| KSR-$L_{21}$ | 0.2726 | 0.2232 | 2999 | 0.3028 | 0.2355 | 2760 | 0.3265 | 0.3046 | 2599 | 0.3938 | 0.2868 | 2265 |
| LPM | 0.3649 | 0.3552 | 1950 | 0.3321 | 0.3089 | 2450 | 0.3334 | 0.3140 | 2324 | 0.3138 | 0.2892 | 2541 |

**Syn360**

| Syn360 | CR=54.1557 | | | CR=81.2641 | | | CR=105.4173 | | | CR=140.4878 | | |
|---|---|---|---|---|---|---|---|---|---|---|---|---|
| | NRMSE | NMAE | NECR | NRMSE | NMAE | NECR | NRMSE | NMAE | NECR | NRMSE | NMAE | NECR |
| CD | 0.1086 | 0.0880 | 20548 | 0.1140 | 0.0927 | 19136 | 0.1195 | 0.0975 | 17838 | 0.1253 | 0.1031 | 16500 |
| FKSC | **0.1024** | **0.0824** | **22255** | **0.1036** | **0.0834** | **22044** | **0.1043** | **0.0839** | **22014** | **0.1048** | **0.0843** | **21711** |
| KSC | **0.1024** | **0.0824** | **22255** | **0.1036** | **0.0834** | **22044** | **0.1043** | **0.0839** | **22014** | **0.1048** | **0.0843** | **21711** |
| KFMC | 0.1401 | 0.1130 | 16127 | 0.1539 | 0.1232 | 14794 | 0.1541 | 0.1244 | 14642 | 0.1398 | 0.1134 | 15768 |
| KSR-$L_{21}$ | 0.1121 | 0.0905 | 19931 | 0.1171 | 0.0941 | 19327 | 0.1076 | 0.0856 | 21947 | 0.1093 | 0.0874 | 21342 |
| LPM | 0.2460 | 0.2030 | 8578 | 0.2528 | 0.2098 | 8082 | 0.2557 | 0.2127 | 8109 | 0.2571 | 0.2161 | 7515 |

**PlanetLab**

| PlanetLab | CR=49.2513 | | | CR=60.4177 | | | CR=68.2878 | | | CR=81.5557 | | |
|---|---|---|---|---|---|---|---|---|---|---|---|---|
| | NRMSE | NMAE | NECR | NRMSE | NMAE | NECR | NRMSE | NMAE | NECR | NRMSE | NMAE | NECR |
| CD | 0.2360 | 0.1597 | 31726 | 0.3221 | 0.2411 | 19466 | 0.2599 | 0.1881 | 23786 | 0.3360 | 0.2543 | 18120 |
| FKSC | **0.2236** | **0.1469** | **36608** | **0.2308** | **0.1575** | 32540 | **0.2216** | **0.1472** | **35106** | **0.2281** | **0.1560** | 32599 |
| KSC | **0.2236** | **0.1469** | 36606 | **0.2308** | **0.1575** | 32540 | **0.2216** | **0.1472** | 35098 | **0.2281** | **0.1560** | 32597 |
| KFMC | 0.2369 | 0.1666 | 32553 | 0.2311 | **0.1557** | **36053** | 0.2417 | 0.1670 | 33887 | 0.2380 | 0.1633 | **34773** |
| KSR-$L_{21}$ | 0.2397 | 0.1633 | 35040 | 0.2365 | 0.1600 | **35159** | 0.2768 | 0.1869 | 33715 | 0.2456 | 0.1643 | **34437** |
| LPM | 0.3563 | 0.2949 | 19655 | 0.3083 | 0.2402 | 24854 | 0.3147 | 0.2499 | 23261 | 0.2999 | 0.2318 | 25542 |

• Advantage of FKSC over other kernel sparse coding methods: In the totality of 20 experimental setups, FKSC fails to achieve the top performance in only 3 instances, signifying its overwhelming dominance in most scenarios. For instance, in the Harvard data at CR = 54.1633, i.e., the data size is compressed to 1.85% of the original scale, the NMAE, NMAE, NECR metrics of FKSC reach 0.2057, 0.1675, 4481 respectively. When KSC is excluded from consideration, KFMC emerges as the sub-optimal method. Compared to KFMC, FKSC exhibits a substantial 30% relative improvement in NRMSE, 36% in NMAE, and 1.36 times boost in NECR. This may be attributed to the solution of FKSC, which boasts a convergence guarantee and facilitates the optimization process in achieving an exceptional stationary point.

## 6 CONCLUSION

In this paper, we have addressed the limitations of traditional sparse coding algorithms when dealing with nonlinear real-world signals. By leveraging RBF kernel to implicitly increase the dimensionality of the original data to enhance its separability, our proposed kernel sparse coding method enables more effective sparse representations and provides a solution with convergence guarantees based on the principle of coordinate descent. To further optimize the computational efficiency, we introduced a novel two-stage acceleration strategy. The strategy is theoretically underpinned by the insight that updates to zero weights can be skipped and empirically supported by the observation that the positions of nonzero weights are relatively stable. This innovation allows the optimization process to be significantly accelerated. Experimental results validate the effectiveness of our approach. The two-stage acceleration strategy demonstrates a remarkable reduction in processing time by up to 90%. Additionally, our method shows superior performance compared to both traditional linear sparse coding methods and other kernel sparse coding techniques, with significantly lower values of NMAE when CR is high.

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

# A APPENDIX

## A.1 RELATED WORK

Sparse coding serves as a pivotal branch within deep learning (Yang et al., 2009), endeavoring to discover a sparse representation of input data in the form of a linear combination of basic atoms. This pursuit achieves multiple objectives: data compression, enhancement of computational efficiency, and uncovering salient features of the data, thereby finding widespread applications across diverse domains.

However, recent studies (Pimentel-Alarcón et al., 2017; Fan et al., 2021; Fan & Udell, 2019) have indicated that many real-world signals do not possess an inherent linear structure, hence traditional linear sparse coding methods may fail to be effective (Yang et al., 2016). Consequently, a few scholars have proposed the use of kernel tricks to implicitly elevate the dimensionality of the input data, thereby enhancing the applicability of sparse coding methods to complex data. Kernel sparse coding is an extension of linear sparse coding. Existing research on kernel sparse coding problems can be roughly divided into two categories based on the form of constraint on the sparse matrix in their mathematical models: the $l_0$ norm-based approach and the $l_1$ norm-based approach.

The kernel sparse coding problem formulated with the $l_0$ norm is inherently NP-hard and can only be solved by using heuristic algorithms. For instance, the Kernel Orthogonal Matching Pursuit (KOMP) (Nguyen et al., 2013) algorithm iteratively selects dictionary atoms that exhibit the highest correlation with the current residual during the sparse optimization process until a predetermined sparsity level is achieved. Alternatively, the Linearized Proximal Method (LPM) (Quan et al., 2016) leverages block coordinate descent, initially obtaining a closed-form solution through the proximal gradient method, followed by a brute-force selection of the weights with the largest absolute values to satisfy the sparsity constraint. These heuristic algorithms are collectively noted for their rapid sparse optimization capabilities. However, **they fall short by lacking convergence guarantees**, which means they cannot assure the attainment of the optimal solution and are susceptible to getting trapped in local optima.

In contrast, the $l_1$ norm offers a convex alternative that is easier to handle mathematically but still presents challenges due to its non-differentiability. The Kernel Feature-sign Search (KFSS) (Gao et al., 2013) algorithm addresses this issue by taking an active approach to guessing the signs of the coefficient weights. This strategic method is based on the observation that the sign of a coefficient is closely related to the correlation between the dictionary atom and the residual. By correctly guessing the signs, KFSS can simplify the optimization problem and work towards finding an analytical solution that satisfies the sparsity requirement. The First-Order Smooth Optimization (FOSO) (Kim, 2014) algorithm tackles the non-differentiability of the $l_1$ norm by approximating it with smooth functions. This approximation allows the use of conventional gradient-based optimization techniques, which are otherwise inapplicable to non-differentiable problems. The smooth approximation serves as a surrogate that enables the derivation of an analytical solution while still promoting sparsity in the solution. Both KFSS and FOSO contribute to the broader field of kernel sparse coding by providing methods that can yield sparse solutions with convergence guarantees. However, **they face challenges in terms of the scalability of optimization complexity and computational speed**. Therefore, this paper aims to explore a new fast kernel sparse coding method with convergence guarantees.

## A.2 KSCDL ALGORITHM

---
**Algorithm 1** KSCDL

**Input:** $\mathbf{Y}, \kappa, \lambda, t_{max}$
1: **Initialize**: $\mathbf{X} \sim \mathcal{N}(0,1), t = 0$
2: **repeat**
3:     $t = t + 1$
4:     Update sparse coding matrix $\mathbf{W}^{(k)}$ through (9)
5:     Update dictionary matrix $\mathbf{X}^{(k)}$ through (11)
6: **until** the convergence condition is satisfied or $t = t_{max}$
**Output:** $\mathbf{W}, \mathbf{X}$

---

The convergence condition in Algorithm 1 refers that the relative error $RE = \|\mathbf{X}^{(k)} - \mathbf{X}^{(k-1)}\|_F / \|\mathbf{X}^{(k-1)}\|_F < \varepsilon$ for any $\varepsilon > 0$. The following Theorem shows this algorithm converges.

**Theorem:** Algorithm 1 converges to a stationary point.

*Proof.* Denote the objective function in (3) as $\mathcal{L}(\mathbf{X}, \mathbf{W})$. On the one hand, when $\mathbf{W}$ is fixed, $\mathcal{L}(\mathbf{X}, \mathbf{W})$ is twice differentiable with respect to $\mathbf{X}$, and the gradient $G_\mathbf{X}$ is Lipschitz continuous with Lipschitz constants $L_\mathbf{X}$, then the Taylor expansion of $\mathcal{L}(\mathbf{X}, \mathbf{W})$ around $\mathbf{X}^{(k-1)}$ is

$$\mathcal{L}(\mathbf{X}^{(k)}, \mathbf{W}) = \mathcal{L}(\mathbf{X}^{(k-1)}, \mathbf{W}) + \langle G_\mathbf{X}, \mathbf{X}^{(k)} - \mathbf{X}^{(k-1)} \rangle \\ + \frac{1}{2} \text{vec}(\mathbf{X}^{(k)} - \mathbf{X}^{(k-1)})^T H_\mathbf{X} \text{vec}(\mathbf{X}^{(k)} - \mathbf{X}^{(k-1)}), \tag{17}$$

where $H_\mathbf{X}$ is the Hessian matrix of $\mathcal{L}(\mathbf{X}, \mathbf{W})$ with respect to $\mathbf{X}$. $\langle \cdot, \cdot \rangle$ is the trace of the inner product of matrices, i.e., $\langle \mathbf{A}, \mathbf{B} \rangle = \text{trace}(\mathbf{A}^T \mathbf{B})$. Substituting (11) into (17) yields

$$\mathcal{L}(\mathbf{X}^{(k)}, \mathbf{W}) \leq \mathcal{L}(\mathbf{X}^{(k-1)}, \mathbf{W}) + \langle G_\mathbf{X}, -\mu G_\mathbf{X} \rangle + \frac{\lambda_{\max}(H_\mathbf{X})}{2} \text{vec}(-\mu G_\mathbf{X})^T \text{vec}(-\mu G_\mathbf{X}) \\ \leq \mathcal{L}(\mathbf{X}^{(k-1)}, \mathbf{W}) - \frac{2\mu - \mu^2 L_\mathbf{X}}{2} \|G_\mathbf{X}\|_F^2, \tag{18}$$

where $\lambda_{\max}(H_\mathbf{X})$ is the maximum eigenvalue of Hessian matrix $H_\mathbf{X}$. Clearly, $\frac{2\mu - \mu^2 L_\mathbf{X}}{2} \geq 0$ if $0 \leq \mu \leq \frac{2}{L_\mathbf{X}}$. That is to say, when $\mathbf{X}$ is updated according to equation (11), the objective function $\mathcal{L}(\mathbf{X}, \mathbf{W})$ is decreasing and bounded below by 0. Hence, the objective function $\mathcal{L}(\mathbf{X}, \mathbf{W})$ with respect to $\mathbf{X}$ is convergent.

On the other hand, when $\mathbf{X}$ is fixed, the kernel sparse coding subproblem (4) constitutes a Lasso problem, with the explicit solution (9) obtained directly through the coordinate descent algorithm. Thus, similar to Fujiwara et al. (2016) and following the convergence analysis presented in Tseng (2001), the objective function $\mathcal{L}(\mathbf{X}, \mathbf{W})$ with respect to $\mathbf{W}$ is convergent. $\square$

### A.3 DISCUSSION ON EXTENDING THE SOLUTION OF SCDL TO HIGHER-DIMENSIONAL FEATURE SPACE

Example 2：Extend the solution of SCDL in the original space to high-dimensional feature space

In Example 1, the SCDL result of input data $\mathbf{Y}$ yields the dictionary matrix $\mathbf{X}$ and the sparse matrix $\mathbf{W}$. In addition, $\mathbf{Y}$ is vertically expanded to high-dimensional data $\widetilde{\mathbf{Y}}$, and the SCDL result of $\widetilde{\mathbf{Y}}$ yields the dictionary $\widetilde{\mathbf{X}}$ and the sparse matrix $\widetilde{\mathbf{W}}$. In this case, the approximation error $AE_{high1} = \frac{1}{2}\|\widetilde{\mathbf{Y}} - \widetilde{\mathbf{X}}\widetilde{\mathbf{W}}\|_F^2 = 0.3625$. In Example 2, suppose we vertically expand the dictionary $\mathbf{X}$ (obtained from Example 1) to dictionary $\overline{\mathbf{X}}$ in the high-dimensional feature space, we can derive the high-dimensional approximate data as $\overline{\mathbf{Y}} = \overline{\mathbf{X}}\mathbf{W}$, where $\mathbf{W}$ is obtained from Example 1. In this case, the approximation error in the high-dimensional feature space is $AE_{high2} = \frac{1}{2}\|\widetilde{\mathbf{Y}} - \overline{\mathbf{Y}}\|_F^2 = 4.8904$, which is considerably higher than $AE_{high1}$. That is to say, when directly extending the results of SCDL in the original space to a high-dimensional feature space, the data approximation effect is not as good as performing SCDL directly in the high-dimensional feature space. In other words, the solutions of (3) should not be obtained by simply extending the solutions of (1). Consequently, it is necessary to explore novel approaches to solve the KSCDL problem (3).

## A.4 FKSC ALGORITHM

The FKSC method with a two-stage acceleration strategy is described in Algorithm 2. The core strategies of FKSC includes: Stage 1: Prioritizing updates of initially predicted nonzero elements (lines 3-11); Stage 2: Safely skipping updates for zero elements in $\mathbf{w}$ (lines 14-16).

**Computational complexity analysis.** Assuming the FKSC method requires $T_1$ rounds of iterations in Stage 1 and $T_2$ rounds of iterations in Stage 2, while the KSC method requires $T$ rounds of iterations for the weight vector to converge.

In Stage 1 of FKSC, assuming that in the 1st round of iteration, there are $k_1$ elements are predicted to be nonzero and require $T_1$ iterations to converge. Thus, the total computational complexity of Stage 1 is $O(p + k_1pT_1)$, where $O(p)$ represents the total computational complexity of calculating the bounds in the 1st round of iteration, and $O(k_1pT_1)$ represents the total computational complexity of the $k_1$ predicted nonzero elements over $T_1$ rounds of iteration.

In Stage 2 of FKSC, assuming that the entire weight vector requires $T_2$ iterations to converge, and on average, $k_0$ elements are predicted to be zero in per round of iteration. Therefore, the total computational complexity of Stage 2 is $O(pT_2(p - k_0 + 1))$, where $O(pT_2)$ is the total computational complexity of calculating the bounds throughout the process, and $O((p - k_0)pT_2)$ is the total computational complexity of the remaining $(p - k_0)$ elements, excluding the predicted $k_0$ zero elements, to undergo $T_2$ rounds of iterations.

Hence, the total computational complexity of the entire FKSC algorithm is $O(p(1 + k_1T_1 + T_2(p - k_0 + 1)))$. If no acceleration strategies are adopted and the standard coordinate descent algorithm is used for updating iterations directly, then the overall computational complexity would be $O(p^2(T_1 + T_2))$. Generally speaking, $T$ is not much different from $T_1 + T_2$, then the two-stage acceleration strategy has the potential to speed up the processing time of kernel sparse coding.

---

**Algorithm 2** Complete organization of FKSC

---

**Input:** $\mathbf{X}, \mathbf{y}, \kappa,$
1: Compute $\kappa(\mathbf{y}, \mathbf{X}), \kappa(\mathbf{X}, \mathbf{X})$
2: **Initialize:** $\mathbf{w}^{(0)} = \mathbf{0}, \mathbf{z}^{(0)} = \mathbf{0}, \Omega = \{1, \cdots, p\}, \widetilde{\Omega} = \emptyset$
3: **for** each $i \in \Omega$ **do**
4:     Compute $\overline{z}_i^{(k)}$ and $\underline{z}_i^{(k)}$ by (14)
5: **end for**
6: **if** $\underline{z}_i^{(k)} > \lambda$ or $\overline{z}_i^{(k)} < -\lambda$ **then**
7:     $\widetilde{\Omega} = \widetilde{\Omega} \cup \{i\}$
8: **end if**
9: **repeat**
10:     Update $w_i^{(k)}$ through (9) for each $i \in \widetilde{\Omega}$
11: **until** $\mathbf{w}$ converges
12: **repeat**
13:     **for** each $i \in \Omega$ **do**
14:         Compute $\overline{z}_i^{(k)}$ and $\underline{z}_i^{(k)}$ by (14)
15:         **if** $-\lambda \leq \underline{z}_i^{(k)} \leq \overline{z}_i^{(k)} \leq \lambda$ **then**
16:             $w_i^{(k)} = 0$
17:         **else**
18:             Update $w_i^{(k)}$ through (9)
19:         **end if**
20:     **end for**
21: **until** $\mathbf{w}$ converges
**Output:** $\mathbf{w}$

---

## A.5 EXPERIMENTAL SETUPS

### A.5.1 DATASETS

The datasets mainly utilized encompass four real-world datasets and one synthetic dataset. Acknowledging the dimensional constraints of the real-world datasets, we specifically introduce a syn-

thetic dataset of an tailored dimension (referred to as "Syn360"), to ensure a progressive scaling of dimensions across the datasets, thereby facilitate our comprehensive observation and analysis of the experimental outcomes.

• **PM2.5** (Zheng et al., 2015) records the air quality data collected by Microsoft Research's Urban Computing team for a year (from May 1, 2014 to April 30, 2015) in the Urban Air project. The dataset covers four major cities in China (Beijing, Tianjin, Guangzhou, and Shenzhen) and 39 adjacent cities within a 300-kilometer radius.

• **Traffic** (Chen et al., 2018) encompasses traffic speed observations from 214 anonymous road segments, primarily comprising urban highways and main thoroughfares, spanning a two-month period from August 1, 2016, to September 30, 2016. The data is recorded every 10 minutes, originating from Guangzhou, China.

• **Harvard** (Ledlie et al., 2007) contains the data of application-level RTT, gathered from interactions among 226 Azureus clients over a span of 72 hours.

• **PlanetLab** (Zhu et al., 2017) consists of RTT measurements between 490 nodes in the PlanetLab network across 18 time slices.

• **Syn360** (similar to (Fan et al., 2021)) is a synthetic data with each column generated by $\mathbf{y} = \psi(\mathbf{s}) = \mathbf{P}\widetilde{\mathbf{s}}$, where $\mathbf{s} = [s_1, s_2, s_3]^T \sim \mathcal{U}(-1, 1)$; $\psi \in \{\psi^1, \cdots, \psi^{15}\}$ is an order-3 polynomial mapping with $\mathbf{P} \in \{\mathbf{P}^1, \cdots, \mathbf{P}^{15}\} \subset R^{360 \times 20} \sim \mathcal{N}(0, 1)$. $\widetilde{\mathbf{s}} = [1, s_1, s_2, s_3, s_1^2, s_1 s_2, s_1 s_3, s_2^2, s_2 s_3, s_3^2, s_1^3, s_1^2 s_2, s_1^2 s_3, s_1 s_2 s_3, s_2^3, s_2^2 s_1, s_2^2 s_3, s_3^3, s_3^3 s_1, s_3^2 s_2]^T \in R^{20}$; For each $\mathbf{P}^i, i = 1, \cdots, 18$, we randomly generate 100 $\mathbf{y}$s.

In Table 3, we present the size of the data selected from each dataset, used for training the dictionary, and for testing the performance of the proposed FKSC algorithm.

Table 3: The processed data size for training and testing

|  | PM2.5 | Traffic | Harvard | Syn360 | PlanetLab |
|---|---|---|---|---|---|
| Training | $174 \times 920$ | $214 \times 1440$ | $226 \times 1130$ | $360 \times 1000$ | $490 \times 1470$ |
| Testing | $174 \times 184$ | $214 \times 288$ | $226 \times 226$ | $360 \times 500$ | $490 \times 490$ |

### A.5.2 COMPETITORS

The comparison algorithms for KSC and FKSC include

• **CD** (Fujiwara et al., 2016): the coordinate descent algorithm for traditional linear sparse coding problem, based on $l_1$ regularization in the original input space.

• **KFMC** (Fan & Udell, 2019): the explicit solution approach for kernelized matrix factorization problems, with the weight matrix regularized by the Frobenius norm.

This method results in a weight matrix that is not strictly sparse, but contains many weight factors that are very small. In this paper, for each column of the weight matrix, we preserve weight factors with significant absolute values while setting the others to zero to meet a predetermined sparsity level, thereby facilitating a comparison with our proposed method.

• **KSR-$L_{2,1}$** (Qian et al., 2023): the explicit solution approach for kernel sparse representation, with the weight matrix being constrained via $L_{2,1}$ matrix norm.

Ideally, this method yields a weight matrix that is row-sparse, but the overall sparsity is not high enough, or in other words, there are not enough zero elements. Therefore, to enable a comparative analysis with our proposed method, for each column of the weight matrix, we retain only those weight factors with substantial absolute values in accordance with a preset sparsity level, while nullifying the others to zero.

• **LPM** (Quan et al., 2016): the linearized proximal method, employed for addressing kernel sparse coding problems with $l_0$ regularization, initiates by leveraging the proximal gradient approach to secure a closed-form solution. Subsequently, it selects and retains the weight factors with the most significant absolute values, aligning with a predefined sparsity level.

### A.5.3 METRICS

The compression performance of an algorithm is primarily reflected in two aspects: compression ratio and accuracy. Therefore, we present the following metrics:

• Compression Ratio (CR):

$$\text{CR} = \frac{\text{Size of original data}}{\text{Size of compressed data}}.$$

• Normalized Root Mean Square Error (NRMSE):

$$\text{NRMSE} = \|\mathbf{Y} - \widetilde{\mathbf{Y}}\|_F \big/ \|\mathbf{Y}\|_F,$$

where $\widetilde{\mathbf{Y}}$ refers to the approximate data obtained by solving the optimization problem (19) for kernel-based algorithms include LPM, KSR-$L_{2,1}$, KFMC, as well as the proposed KSC and FKSC, and $\widetilde{Y} = \mathbf{XW}$ for sparse coding algorithms in the original space like CD.

• Normalized Mean Absolute Error (NMAE):

$$\text{NMAE} = \sum_{(i,j)} |y_{ij} - \widetilde{y}_{ij}| \big/ \sum_{(i,j)} |y_{ij}|,$$

where $\widetilde{y}_{ij}$ is the element located at the $i$th row and $j$th column of matrix $\widetilde{\mathbf{Y}}$.

• Number of Elements Correctly Reconstructed (NECR):

$$\text{NECR} = \sum_{(i,j)} y_{ij}^{\varrho}, \quad y_{ij}^{\varrho} = \left\{ \begin{array}{ll} 1, & |y_{ij} - \widetilde{y}_{ij}| < \varrho; \\ 0, & \text{Otherwise,} \end{array} \right.$$

where $\widetilde{y}_{ij}$ is the element located at the $i$th row and $j$th column of matrix $\widetilde{\mathbf{Y}}$ and $\varrho = 10^{-3}$.

In these metrics, higher values of CR and NECR indicate superior compression performance by the algorithm. Conversely, lower values of NRMSE and NMAE signify a more effective compression capability.

### A.6 EXPERIMENTAL RESULTS

For all experiments, the maximum number of iterations is set to 100, and the algorithm termination tolerance is set to $1e - 3$. For KSCDL, the termination tolerance refers to the relative error $RE_{\mathbf{X}} = \|\mathbf{X}^{(k)} - \mathbf{X}^{(k-1)}\|_F / \|\mathbf{X}^{(k-1)}\|_F$ , while for FKSC, the termination tolerance refers to the relative error $RE_{\mathbf{W}} = \|\mathbf{W}^{(k)} - \mathbf{W}^{(k-1)}\|_F / \|\mathbf{W}^{(k-1)}\|_F$.

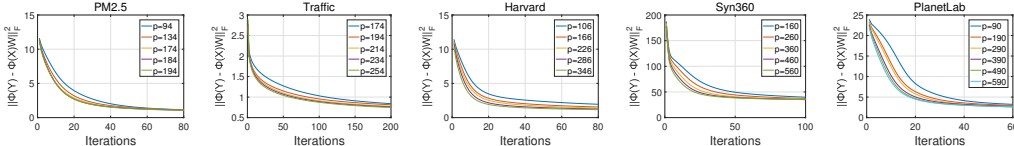

Figure 10: The objective value for different numbers of atoms (i.e., $p$), with $\lambda = 0.001$ and $\sigma = 2$.

**Convergence Verification:** To validate the convergence of the KSCDL method outlined in Algorithm 1, we set $\lambda = 0.001$ and $\sigma = 2$. As shown in Fig. 10, the objective values $\|\phi(\mathbf{Y}) - \phi(\mathbf{X})\mathbf{W}\|_F^2$ achieve convergence across the evaluated datasets, even as the number of atoms varies. It is noteworthy that the objective value for Syn360 is comparatively higher than those of the other datasets. This discrepancy arises from the less optimal parameter setting of $\sigma = 2$ for this dataset. Further experimentation demonstrate that adopting a value of $\sigma$ exceeding 10 leads to significantly enhanced performance.

In addition, we can also observe that as the number of atoms increases, the convergence speed of the objective function gradually accelerates. Nonetheless, once the number of atoms surpasses the number of rows ($174, 214, 226, 360$ and $490$ for PM2.5, Traffic, Harvard, Syn360 and PlanetLab,

respectively) in the dictionary matrix, the convergence behavior of the objective value is no longer sensitive to the number of atoms. Therefore, setting the number of atoms in the dictionary equal to the number of its rows, that is, configuring the dictionary as a square matrix, is reasonable.

**Hyperparameter Analysis:** In addition to the number of the dictionary atoms, there are also two hyperparameters that can influence the outcomes of the KSCDL algorithm: the sparsity penalty parameter $\lambda$ and the RBF kernel hyperparameter $\sigma$. With all dictionaries configured as square matrices, Fig. 11 indicates that when $\lambda$ is fixed and $\sigma$ is sufficiently large, the objective value remains insensitive to variations in $\sigma$. Conversely, when $\sigma$ is kept constant, the objective function exhibits a tendency of exponential growth as $\lambda$ increases. To ensure that the objective value of KSCDL remains within a desirable range, we will restrict the value of $\lambda$ within the interval of $[0.001, 0.1]$, from this point onward.

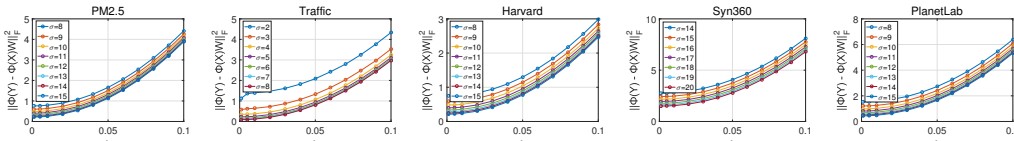

Figure 11: The influence of different $\lambda$ and $\sigma$ on objective value.

## A.7 Application of FKSC

FKSC transforms dense input data into a sparse representation, thereby enabling solutions for challenges such as data compression, improving of computational efficiency, feature analysis and denoising. Taking data compression as an example, Fig. 12 delineates the application procedure of FKSC. Initially, a dictionary matrix $\mathbf{X}$, which can reflect the underlying structural features of the data, is derived from historical data using the KSCDL algorithm outlined in Algorithm 1, with KSC being replaced by

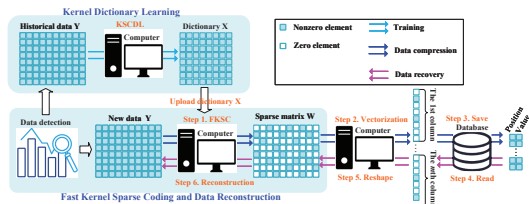

Figure 12: Example for FKSC's application.

FKSC. In the subsequent stage, the original data undergoes a three-step compression process (i.e., Step1-3) utilizing this dictionary matrix $\mathbf{X}$. Conversely, the compressed data can be approximately reconstructed into its original form through a three-step decompression procedure (i.e., Step4-6). The specific implementation details of each step are outlined below:

• Step 1: Encode new data $\mathbf{Y}$ into sparse matrix $\mathbf{W}$ via the propose FKSC method, leveraging the dictionary matrix $\mathbf{X}$ obtained from the initial stage.

• Step 2: Vectorize the sparse matrix $\mathbf{W}$ to facilitate its subsequent compression.

• Step 3: Save the vectorized $\mathbf{W}$ in a condensed two-column matrix. Column one lists indices of non-zero elements, and column two lists their values.

• Step 4: Decode back the small matrix from the storage unit into a sparse column vector.

• Step 5: Fold the sparse column vector back into the sparse matrix $\mathbf{W}$.

• Step 6: Solve the following optimization problem to obtain the approximate data $\widetilde{\mathbf{Y}}$:

$$\widetilde{\mathbf{Y}} = \arg\min_{\widetilde{\mathbf{Y}}} \frac{1}{2}\|\phi(\widetilde{\mathbf{Y}}) - \phi(\mathbf{X})\mathbf{W}\|_F^2. \tag{19}$$

