# OpenReview forum: "A Novel Kernel Sparse Coding Method with A Two-stage Acceleration Strategy"
_ICLR.cc/2025/Conference — ICLR 2025 Conference Withdrawn Submission_

### Official Review · Reviewer_jtLb · 2024-10-27

**Soundness:** 1
**Presentation:** 1
**Contribution:** 1
**Rating:** 1
**Confidence:** 5

**Summary:**

The paper addresses sparse coding and dictionary learning with a kernel-based model, with a focus on the RBF kernel. A two-stage acceleration strategy is also proposed.

**Strengths:**

maybe

**Weaknesses:**

There are many major issues in the paper, making the paper not suitable for publication. I provide in the following some of the major issues.

A major issue is the low novelty of this work. The authors present their kernel-based model as if it is the first time that sparse coding and dictionary learning is considered in the nonlinear kernel-based model. This is a pity, as the kernel-based embedding for nonlinear sparse coding and dictionary learning has been largely investigated in the literature in the 15 or even 20 years.

The authors fail to position their contributions in such a literature.

Moreover, the first stated contribution of this paper is, as described by the authors, the fact that an RBF kernel maps the data into an infinite dimension space. This has been known since the end of last century, and is central in kernel-based models and the much old support vector machines. Expressions as (2) and following discussions have been well known in the literature.

There are many issues with the notations, as described in the following:

phi(.) is defined on what space ? if defined on the vector space (which should be the case in this paper), then phi(y) is relevant, but not phi(Y) because Y is a matrix. For the same reasons, phi(X) is not correct.

What is the dimension of phi(.) ? In this end of section 2, it is argued that it is an infinite dimension because of the RBF kernel. Therefore, the Frobenius norm in (3) is not correct, as it is not correct that phi(X) is multiplied by a matrix W of infinite dimension !
Moreover, this norm is transformed to the L2-norm in (4), which is also not correct. Furthermore, all the following Frobenius and L2-norms are not correct. This means that the computation of the Trace given in (10) is not correct.

**Questions:**

No questions.

---

### Official Review · Reviewer_mN83 · 2024-11-03

**Soundness:** 2
**Presentation:** 2
**Contribution:** 2
**Rating:** 3
**Confidence:** 4

**Summary:**

This paper focuses on the sparse coding setting and extends the linear setting to non-linear setting with the Radial Basis Function kernel (c.f. Equation (3)).
To solve the problem, they propose to use the alternative minimization method, that is, decompose the problem into two sub-questions, i.e., Kernel Sparse Coding (KSC) subproblem and Kernel Dictionary Learning (KDL) sub-problem. Then, they propose two acceleration scheme, which is safely skipping update for zero elements and prioritizing updates for initially predicted nonzero elements.

**Strengths:**

This paper is easy to understand.  The extension from the linear scheme to non-linear scheme is quite natural.

**Weaknesses:**

1. The novelty is limited. Basically, the proposed method is to replace the original $Y$ and $X$ with $\phi(Y)$ and $\phi(X)$. The alternative minimization method to update $X$ and $W$ is quite standard. For example, minimizing $W$ (c.f. Eq (7)) is the same as the Lasso estimator. The solution in (8) and (9) is called the soft-thresholding estimator, which is standard in solving the Lasso algorithm. The solution of $X$ is found by gradient descent algorithm. Although the acceleration scheme is proposed, no theoretical analysis, say the convergence is provided.
2. The baseline to prove the acceleration is their own algorithm, which is not well accepted and may lead to question of the comparison fairness.

**Questions:**

1. In Line 411, it claims that the CR is determined by FKSC. Why this setting is fair to other algorithms?
2. On what criteria were the comparison algorithms selected?
3. In Line 160, you have $|q| = 0$, can you explain more. If all entries $q_i$ are non-negative, we have all entries to be identically zero. Otherwise, we have some entries being negative. This leads to questions regarding the definition of $q_i!$. I understand there is still definition on the factorial of negative numbers but am still a little confused.

---

### Official Review · Reviewer_1cpX · 2024-11-03

**Soundness:** 2
**Presentation:** 3
**Contribution:** 2
**Rating:** 5
**Confidence:** 4

**Summary:**

This paper proposes a kernel sparse coding method using the Radial Basis Function (RBF) kernel to handle nonlinear data. The proposed approach employs l1-norm regularization and coordinate descent to guarantee convergence. Two acceleration strategies are developed to improve computational efficiency. However, the contribution is not very clear, with not enough evidence showing that the proposed method performs better than existing KSC methods.

**Strengths:**

The problem formulation, theoretical analysis, and acceleration strategies are technically sound.

**Weaknesses:**

My main concern is that KSC using RBF has already been studied, such as KSR, KFSS, and FOSO (the author has mentioned these works ).  However, the author doesn't mention them in the introduction, making it confusing that using RBF for KSC is a new method. Further, there are not enough comparisons between the proposed method and existing KSC methods (such as KFSS and FOSO) in the experimental results. So it is quite unclear what the main contribution of the proposed method is. (The acceleration strategies might be the contribution but the overall contribution should not be focused on using RBF for KSC.)

**Questions:**

1. It is recommended to add a paragraph in the introduction summarizing previous RBF-based KSC approaches like KSR, KFSS, and FOSO, rather than put them just in the appendix. Also, please clearly state how the proposed method differs from or improves upon the existing methods.
2. Please add additional experimental results in Table 2 showing the performance of the proposed method compared with KFSS and FOSO. Also, please give some discussions about the comparison results.
3. How about the time cost in real-world data? Please add the corresponding running times for Table 2.
3. Please revise the main contributions of the proposed method. It is needed to demonstrate that this work has enough novelty compared with existing RBF-based KSC approaches like KFSS and FOSO.

---

### Official Review · Reviewer_66T7 · 2024-11-03

**Soundness:** 2
**Presentation:** 3
**Contribution:** 2
**Rating:** 5
**Confidence:** 4

**Summary:**

The authors claim that they proposed a kernel-based sparse coding method and proposed a fast solver for that.

**Strengths:**

The paper is well written

**Weaknesses:**

The Kernel-based sparse coding and subspace sparse clustering exist in the literature. Also using the subgradient method for the solution of the problems with  l1 norm is the primary method. So only contribution of the paper seems to be section 4, which in my belief is not an important contribution.

**Questions:**

Is the kernel sparse coding is new idea?
why use subgradient?

---

### Note · Authors · 2024-11-23

I have read and agree with the venue's withdrawal policy on behalf of myself and my co-authors.